# Long-term Intracortical Neural activity and Kinematics (LINK): An intracortical neural dataset for chronic brain-machine interfaces, neuroscience, and machine learning

**Hisham Temmar[1], Yixuan Wang[2], Nina Gill[3], Nicholas B. Mellon[4], Chang Liu[4],**
**Luis Hernan Cubillos[5], Rio I. Parsons[1], Joseph T. Costello[4,6], Matteo Ceradini[8],**
**Madison M. Kelberman[1], Matthew J. Mender[1,6], Aren I. Hite[1], Dylan M. Wallace[5]**
**Samuel R. Nason-Tomaszewski[1,*] Matthew S. Willsey[1,6], Parag G. Patil[1,6],**
**Anne W. Draelos[1,7], Cynthia A. Chestek[1,5]**

[1]Biomedical Eng., [2]Psychology, [3]Neuroscience, [4]Electrical Engineering and
Computer Science, [5]Robotics, [6]Neurosurgery, [7]Computational Medicine
& Bioinformatics, University of Michigan, Ann Arbor, MI, USA.
[8]The Biorobotics Institute, Sant'Anna School of Advanced Studies, Pisa, Italy.

## Abstract

Intracortical brain-machine interfaces (iBMIs) have enabled movement and speech in people living with paralysis by using neural data to decode behaviors in real-time. However, intracortical neural recordings exhibit significant instabilities over time, which poses problems for iBMIs, neuroscience, and machine learning. For iBMIs, neural instabilities require frequent decoder recalibration to maintain high performance, a critical bottleneck for real-world translation. Several approaches have been developed to address this issue, and the field has recognized the need for standardized datasets on which to compare them, but no standard dataset exists for evaluation over year-long timescales. In neuroscience, a growing body of research attempts to elucidate the latent computations performed by populations of neurons. Nonstationarity in neural recordings imposes significant challenges to the design of these studies, so a dataset containing recordings over large time spans would improve methods to account for instabilities. In machine learning, continuous domain adaptation of temporal data is an area of active research, and a dataset containing shift distributions on long time scales would be beneficial to researchers. To address these gaps, we present the LINK Dataset (Long-term Intracortical Neural activity and Kinematics), which contains intracortical spiking activity and kinematic data from 312 sessions of a non-human primate performing a dexterous, 2 degree-of-freedom finger movement task, spanning 1,242 days. We also present longitudinal analyses of the dataset's neural spiking activity and its relationship to kinematics, as well as overall decoding performance using linear and neural network models. The LINK dataset and code are freely available to the public through the dataset website (`https://chesteklab.github.io/LINK_dataset/`).

---

[*]Currently: Wallace H. Coulter Dept. of Biomedical Engineering, Emory University and Georgia Institute of Technology, Atlanta, GA, USA

39th Conference on Neural Information Processing Systems (NeurIPS 2025) Track on Datasets and Benchmarks.

# 1 Introduction

Due to their limited ability to independently engage in activities of daily living, people living with paralysis often report feelings of isolation and lack of personal agency [1–3]. Brain-machine interfaces (BMIs) aim to restore this ability by using recordings from intact neural circuits in the brain and 'decoding' user intent to control an end effector. BMIs have enabled humans and nonhuman primates (NHPs) to control computer cursors, robotic arms, individuated fingers on a virtual hand, and their own (paralyzed) arms through electrical stimulation [4–17]. Additionally, BMIs have helped restore speech in people living with ALS using recurrent-neural network decoders [18–20]. Intracortical brain-machine interfaces (iBMIs) in particular have emerged as a highly promising avenue, due to the high specificity of spiking activity they capture.

However, the specificity of iBMI recordings also leads to one of the primary barriers for real-world translation: instability of neural signals over time. While semi-stable over very short timescales [21–25], intracortical recordings exhibit significant non-stationarities on longer ones [24, 26], due to factors including the foreign body response, scarring, neuronal death, and constant micro-motions in the brain [27–30]. Because of these shifts, current iBMI decoders require frequent recalibration on supervised data, which typically involved a disruptive and lengthy training session. Many approaches have been proposed for addressing this problem, typically aiming to limit the time spent recalibrating or eliminate recalibration altogether through continual unsupervised alignment of neural data [31–42]. Many have recognized the need for standardized benchmarks on which to compare such methods, such as the FALCON dataset [43]. While FALCON covers a broad variety of tasks in the BMI space, the amount of data per task is limited and the timespan is limited to months, not years.

Adjacent to iBMIs, a growing body of neuroscience research uses intracortical recordings to study the dynamics of neural populations, often in the context of motor or cognitive behavior. These studies apply dimensionality reduction and latent variable models to uncover how neural populations perform the computations driving behavior [44–48]. Multiple studies have shown that this latent structure remains stable on a population-level over time, despite recording from unstable individual units [49, 50, 45]. This has led to the hypothesis that behaviorally-relevant computations are embedded in stable manifolds, which persist over time despite turnover at the level of single neurons [40, 49, 39]. However, evidence remains sparse over long timescales, and questions remain about the degree to which these manifolds evolve with experience or become harder to observe with array degradation.

In machine learning, research on domain adaptation for timeseries forecasting generalizes the issue observed in neural decoding to any non-stationary signal over time. Typical forecasting models can be limited by the amount of data available for training, and their training is further complicated by when the underlying distributions of input features and output spaces evolve over time. Domain adaptation methods seek to resolve this problem of changing data distributions [51–53]. For timeseries data, domain adaptation is additionally challenging due to the temporal nature of the data, where patterns learned on an early section of data are not useful for prediction on a later portion. Neural decoders capable of adapting to domain changes in neural and/or kinematic data can indeed be more robust to performance degradation across time [54, 38]. Yet continuous domain adaptation methods for neural network models and times series data have not yet been widely applied, in part due to the lack of available data sampled across both short and long timescales.

To address these needs, we present the Long-term Intracortical Neural activity and Kinematics Dataset (LINK), which contains 312 sessions on 303 days, spanning ~3.5 years of a single non-human primate performing a trial-based dexterous finger movement task. The dataset contains two pre-processed neural features for use in neural decoding tasks and population-level analyses of neural dynamics over time. We hope it can serve as a test bed for developing new domain adaptation approaches for timeseries data more generally. In this work, we describe the dataset and present initial analyses on the quality of neural signals, their relationships with behavior, and some iBMI decoding results.

# 2 Related Work

Several publicly available datasets have supported progress in BMI decoder performance and robustness, as well as neural population dynamics research. The Neural Latents Benchmark (NLB) [55] introduced a dataset with good task variability, including motor and cognitive tasks, but its temporal span is limited, with most tasks containing single sessions. The more recent FALCON

benchmark [43] includes a wide breadth of tasks and subjects, including data from birds, monkeys, and humans performing multiple motor tasks. However, it is limited in its temporal span and has a sparser distribution of days, making it difficult to study long-term stability.

Beyond benchmark datasets, other relevant open-access datasets have been released through scientific publications. These range from single-session data, such as Brochier et al. [56], to longer spans with lower resolution, such as Gallego-Carracedo et al. [57], which includes data from one subject over 540 days, but only 10 recording sessions. Some studies have examined decoder alignment and stability over extended periods, but the data are either not open-access [49, 40] or cover shorter timescales [58, 18]. LINK fills a key gap by providing long-term, open-access, preprocessed neural and kinematic data with high trial counts, suitable for both decoder stability studies and broader neuroscience research.

## 3 The LINK Dataset

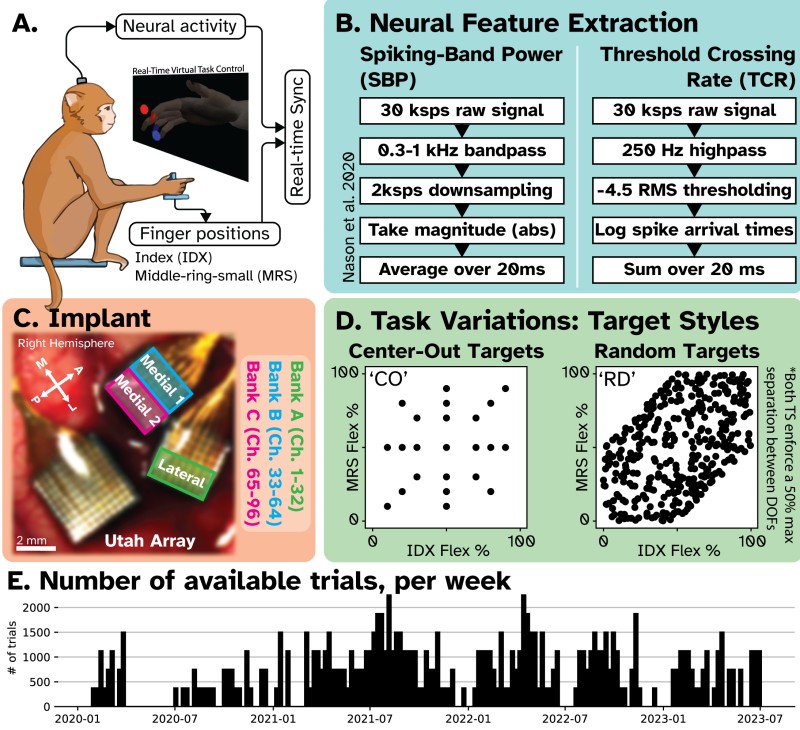

Figure 1: **Overview of Experimental Setup. A.** A non-human primate (Monkey N) performed a 2 degree-of-freedom (DOF) finger movement task where he moved his fingers to match targets presented over a virtual hand on a screen. We simultaneously recorded neural activity from 96 channels of a Utah microelectrode array and the positions (using flex sensors) of the two finger groups, index and middle-ring-small. Note that the illustration depicts Monkey N performing the task with his right hand, but in reality he performed the task with his left. **B.** In this dataset, we provide two standard neural features used in iBMI decoding, spiking-band power and threshold crossing rates. The process for extracting each feature (per channel) is shown here. **C.** Monkey N was implanted with 3 Utah arrays, 2 of which were implanted in the motor cortex. Given available hardware, we were able to record 96 channels simultaneously, specified by the colored rectangles. **D.** We included two variations to the task in this dataset, determined by the pattern of targets presented. The first is 'center-out' which enforces stereotyped movements 'out' with every other trial being a return to 'rest' (the 50% flexion and extension dot) and the second is 'random', where targets are randomly presented but with a maximum 50% separation of the fingers. **E.** A histogram showing the number of available trials per week over the 1,242 days contained in the dataset.

The **L**ong-term **I**ntracortical **N**eural and **K**inematics (LINK) Dataset contains data from a single non-human primate, Monkey N, performing a trial-based, 2 degree-of-freedom (DOF) finger task, recorded

on 303 days spanning 1,242 days. The dataset comprises 312 'sessions' of 375 (mostly) contiguous trial acquisitions. Each session contains one of two variations to the task, 'center-out' or 'random', described below. On 9 days, two sessions are available as both tasks were performed. Monkey N was implanted with three microelectrode arrays, two in the right motor cortex and one in the sensory cortex. While Monkey N performed the task, we synchronously measured the spiking activity of neurons from 96 motor array electrodes and the kinematics of the 2DOF task. In each session, we provide neural data preprocessed into two neural features in 20ms bins: spiking-band power (SBP) and threshold crossing rate (TCR). We also provide the position and velocity of the 2DOF at the same resolution. The dataset also includes trial and session-specific information, like the date of acquisition, target pattern used for the session, target positions per trial, and trial timestamps. Each session is included as a separate file conforming to the Neurodata without Borders (NWB) standard, which can be loaded using the pyNWB or MatNWB APIs. The dataset is publicly available on the DANDI archive, `dandiarchive.org/dandiset/001201`, and code for loading the dataset and replicating all analyses in this paper is available at `github.com/chesteklab/LINK_dataset`. Additionally, the dataset and code can be accessed through the landing page at `https://chesteklab.github.io/LINK_dataset/`.

## 3.1 Experimental Setup

**Behavioral Task.** A nonhuman primate (NHP, Macaca mulatta), Monkey N, was trained to sit in a chair and perform a trial-based, two DOF dexterous finger movement task, shown in Figure 1A. Monkey N's left hand was positioned in a manipulandum, in which he moved two 'finger groups', the index (IDX) and middle-ring-small (MRS) fingers, to control the positions of matching fingers of a virtual hand present on a screen in front of him. The positions of these finger groups along their respective movement arcs were measured using flex sensors calibrated so that a 0 reading was full extension and 1 was full flexion for each DOF. The trial-based task was as follows: Monkey N was presented with two spherical targets at the beginning of each trial, and had to move his fingers so that the virtual finger groups were in each target (see Figure 1A). After holding each finger group in the targets for 750 ms, the trial was considered a success, and he received a juice reward. If unable to acquire targets after a set time (typically 10 seconds), the trial was considered a failure, and the next set of targets was shown. In this dataset, only successful trials are included. Trials were performed in continuous blocks, called 'sessions'. During a session, targets were presented in different patterns of succession, referred to as 'target styles'. The LINK dataset contains two target styles, *'center-out'* (CO), which mimics the center-out-and-back pattern described in previous studies [59, 60], and *'random'* (RD), in which targets were pseudo-randomly chosen. Both are shown in Figure 1D. In both cases, the range of movement for split targets (e.g., IDX flex, MRS extended) was limited to 50% of the full movement arc, as Monkey N could not reliably extend past this due to natural dexterity. Please see Appendix A for more details on the behavioral task.

**Hardware and Feature Extraction.** Monkey N was implanted with 3 Utah microelectrode arrays (Blackrock Neurotech, Salt Lake City, UT, USA). One $10 \times 10$ array was implanted in the somatosensory cortex, and two $8 \times 8$ arrays were implanted in the hand area of the motor cortex based on anatomical targets, shown in Figure 1C and previously described in [61, 62]. The protocols in this study were approved by the Institutional Animal Care and Use Committee at the University of Michigan. Due to hardware limitations, the dataset contains a subset of 96 out of the 128 available motor cortex channels and no somatosensory channels. These were recorded in 32-channel banks, indicated in Figure 1C, labeled by their relative laterality along the motor cortex. For further details on the implants, refer to Appendix A. Neural activity was recorded at 30 kHz by a Cerebus Neural Signal Processor (Blackrock Neurotech), which then extracted two neural features per channel: spiking-band power (SBP) and threshold crossings (TC), outlined in Figure 1B and described in Appendix A. Spiking-band power applies a 300-1000 Hz bandpass filter to each channel, a band which contains the majority of spiking activity, as shown in [63–65]. Threshold crossings are measured by recording time points at which a channel's voltage passes a $-4.5 \times RMS$ (root mean square) threshold set per-channel per day. Both of these features are then binned (averaging for SBP, summing for TCs) into 20ms bins. These binned features will be referred to as SBP and threshold crossing rate (TCR). The recordings included in the dataset begin on day 349 post-implant, up to 1591 days post-implant.

## 3.2 Data Selection

Over the ~3.5 years that Monkey N's arrays were active and usable for real-time decoding, he typically performed the 2-DOF task 5 days a week (excluding holidays and breaks). However, many were unsuitable for this dataset due to factors like task variation, modification to recording paradigms, high noise, or poor NHP behavior (e.g., low motivation that day). To limit external variability, we performed a three-stage dataset curation process: First, we reviewed all experimental notes for Monkey N and identified candidate blocks of contiguous trials (sessions) for each day, if any were present. Second, we loaded every candidate session and filtered out unsuccessful trials, trials with the wrong target style, closed-loop trials, and trials whose hold times were not 750 ms. To prevent bias in decoder training due to uneven numbers of trials across days, we capped the number of trials to 375 per available target style per day. At this point, we also extracted and binned the neural, kinematic, and trial data. Finally, we conducted a full manual review of each session, inspecting neural data for artifacts, baseline shifts, and other issues, excluding sessions with unsuitable data. These stages were repeated as needed until we settled on a final version of the dataset. For details on each stage of the process, please see Appendix B.

## 3.3 Data Format and Contents

Each session is contained in an HDF5 file compliant with the Neurodata Without Borders (NWB, [66]) standard. NWB aims to define a common data standard to allow for collective development of tools for neurophysiological data. In each file, we provide the timeseries data (SBP, TCR, behavior, and experiment time) and relevant trial information (target positions, trial start times, lengths, target style). We also provide mappings of the channels to their locations on each array, along with approximate impedance measurements per channel when available. A visual diagram of the channel mappings to electrode positions is available in Figure S1. For ease of use, we have also included code in the dataset repository that converts the NWB files into simple Python dictionaries. Data and associated metadata were organized into a BIDS-like ([67]) structure and uploaded to the DANDI archive ([RRID:SCR_017571], `https://dandiarchive.org/dandiset/001201`), using the Python command line tool `dandi-cli`. Further details on data format and contents are included in Appendix B.

# 4 Analyses

Here we present several analyses that highlight potential applications of the LINK dataset. Please refer to Appendix C for detailed descriptions of the methods used in these analyses.

## 4.1 Characterizing Neural Activity Over Time

To our knowledge, this is the largest and time-dense set of neural recordings from a Utah array ever released. As such, we aimed to describe how the neural data evolved over the lifespan of the array, irrespective of behavior. We started by measuring the average SBP of each channel, for each session, shown in Figure 2A. Average SBP across channels decreased gradually by approximately 0.033 $\mu V$ per month, as estimated by a linear model. There are periods where groups of channels exhibit irregular shifts (e.g. ~2022-07 to ~2022-11), but these did not affect overall decoding performance. During iBMI experiments, we can roughly estimate the number of active channels by counting those with mean TCR > 1 Hz, shown in Figure 2B. Note that this appears to decrease over time. We then used the SBP of these active channels to calculate participation ratio (PR) per day, shown in Figure 2C. Participation ratio roughly corresponds to the number of dimensions required to explain 80% of the total variance in the neural population, as described in [68]. Generally, 10 dimensions were sufficient to capture the majority of the variance in active channels, and a linear model estimated that PR decreased by ~0.07 PR per month. In Figures S4 and S5, we grouped average SBP of each channel by electrode location (32 channel banks) and by active/inactive channels. We observed lower overall activity on the 'lateral' channels and lower SBP in 'inactive' channels over the course of the dataset. All categories saw steady decreases in average SBP over time.

Additionally, we investigated the evolution of neural activity at a population level. We first concatenated the SBP across all sessions, z-scored per channel, and fit a PCA transform across the entire dataset. Then, we measured the per-day centroid of the top 3 PCs and plotted them in Figure 2D.

We also measured the centroids when grouping the data per-quarter (of a year), shown in larger points, and plotted the standard deviation of these larger groupings as shaded ellipsoids. Visually, these centroids appear to move throughout the PCA space over time, and except for two quarters, the standard deviation of the top 3 PCs does not exhibit large changes.

We then examined the behavior of the neural population during different movements. To do this, we fit PCA transforms to per-channel z-scored neural data within each day, and grouped trials by year and required movement direction (as shown in Figure 2E). Trajectories through the PC space were then averaged over time-aligned trials (by max jerk pre-movement onset) within each year and movement direction. This was performed over both center-out and random trials. The average trajectories for 2020 and 2023 (up to the end of the dataset, containing ~6 months) are shown in Figure 2E. The trajectories shown remain visually separable at the beginning and end of the dataset. For further details about how we examined neural activity over time, please refer to Appendix C.3.

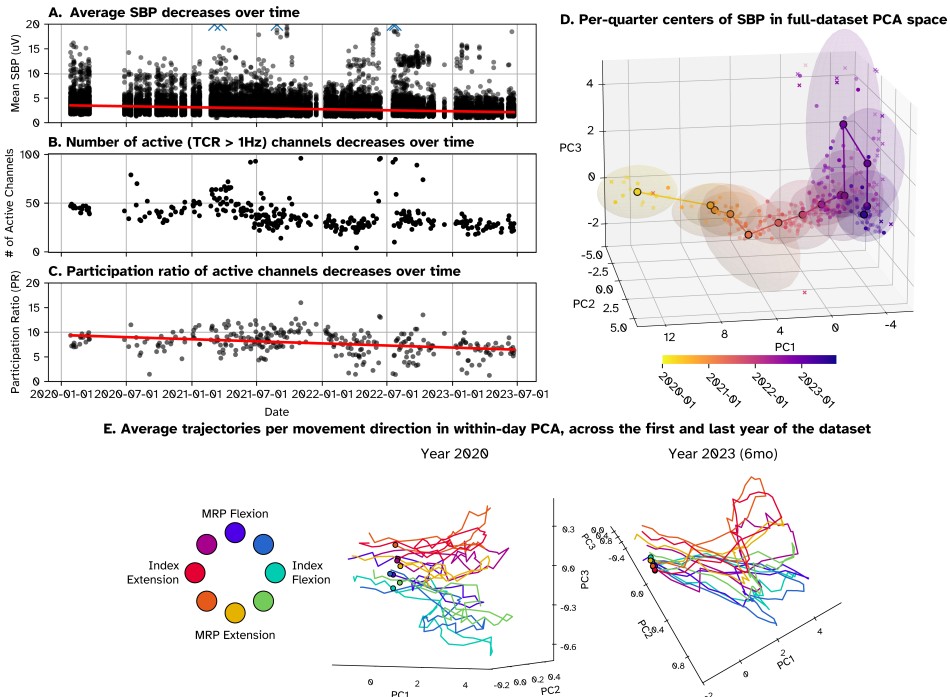

Figure 2: **Changes in neural signals over time.** Here we investigated the change in neural activity over time using various metrics. Additionally, we briefly examine how low-dimensional representations of population-level neural activity relates to behavior of the course of the dataset. **A.** In this plot, each point represents the average SBP of a single channel on a single day. In red, a linear model shows the change in average SBP across channels over time (slope 0.033 $\mu V$ per month). For ease of visualization, the y-axis is limited, but outliers are marked with blue X's. **B.** We counted the number of 'active' channels each day with an average TCR of >1 Hz. This plot shows the number of active channels on each day included in the dataset. **C.** We measured the participation ratio using the SBP of all 'active' channels for each day and plot them over time. A linear model was fit to the data, shown in red. **D.** We z-scored SBP per-channel and fit a PCA transformation across the entire dataset. The mean of the top 3 PCs per day (small dots) are shown as the small dots, colored by date as shown in the colorbar. The mean of the top 3 PCs per-quarter (of a year) are shown as the larger dots, also colored by date. The shaded ellipsoids represent the standard deviation of the top 3 PC scores grouped by quarter. For visual clarity, axis limits were restricted to what is shown, but any outliers are represented with X's at the axis limits, colored by date. **E.** On each day, SBP was z-scored per-channel and a PCA transform was applied. Trials were then grouped by general direction of required movement and aligned by max jerk before movement onset. The two plots show the average trajectories in the top 3 PCs for each movement direction in the first and last year.

## 4.2 Preferred Tuning Direction

One approach to investigating how the relationship between neural activity and behavior changes over time is via the individual relationship between a channel and each degree of freedom. We calculated the *preferred tuning vector* of each channel for each session by fitting independent linear regressions between each channel and the position of both degrees of freedom, and taking the weights of each as a vector, as described in Appendix C.2. These vectors are then decomposed into a 'tuning direction' and 'tuning strength', as shown in Figure 3A. Figure 3B visualizes the preferred tunings of two channels over the entire dataset, one point for each day colored by date. Both channels were considered active, with average TCRs >1 Hz across the dataset. Channel 7 was strongly tuned to MRS flexion throughout the dataset, whereas channel 32 exhibited large shifts in tuning angle. The preferred tuning angles and strengths of all channels on all days are shown in Figures 3C and D, respectively. Per-channel tuning angles and strengths generally appear consistent throughout the dataset; however, many channels have very low tuning strengths throughout the lifespan of the arrays. Such channels also typically showed low average TCRs per day, as shown in Figure S3. To quantify the range of tuning angles per channel across time, we measured the circular median and circular IQR (inter-quartile range) of tuning angle per channel over the whole dataset and the median and IQR of tuning strength per channel over the whole dataset (Figure 3E, described in Appendix C.2). From these plots, it appears that channels in the medial array (see Figure 1) are mainly tuned to MRS flexion, with some IDX flexion, while channels in the lateral array are more tuned to extension of both DOF. In most channels, the circular IQR of tuning angles was small, suggesting their tunings did not change much throughout the dataset.

## 4.3 Neural Decoding

To validate each session and corresponding decoders, each session was split into 300 trials of training data and 75 hold-out trials. The SBP of the training data was z-scored and used to train a ridge regression (RR) and a long short-term memory network (LSTM, [69]). More details on decoder training can be found in Appendix C.1. These decoders predicted both position and velocity of the two finger groups, although in most BMI applications, only velocity is used for closed-loop control. Prediction accuracy ($R^2$) across DOFs for each day can be seen in Figure 4A. Within-day decoding accuracy remains relatively stable across time, and the majority of shifts in accuracy are reflected in both decoders, suggesting these shifts may be due to changes in modulation strength highlighted in Figure 3B. Prediction accuracy by DOF (and position and velocity) is shown in Figure 4B. In general, the position accuracies of the two finger groups outperformed velocities, and MRS predictions outperformed IDX predictions, suggesting there may be more information regarding the MRS finger group in the neural data than IDX.

To visualize the problem of decoder failure over time, all decoders were evaluated on all holdouts within 100 days of training. All decoder predictions were grouped by the number of days out from decoder training, starting with day 0 (decoder training). Figure 4D shows average accuracy across these groups. Decoding accuracy across DOFs was averaged over all predictions. Each performance was labeled with the dates of decoder training, hold-out sets, and the number of days from training. LSTM outperformed RR on all days, but both followed a similar decay curve.

Many BCI decoding algorithms use multiple days of pre-training to improve performance. To test this, we trained LSTMs on up to 5 consecutive session and evaluated their performance on all sessions within 180 days of day 0 (latest day). We excluded days with multiple sessions and the first 10 sessions of the dataset. Note that since sessions are not evenly distributed the time span covered by training sessions varies. Performances ($R^2$) were grouped by relative day of prediction (day k from training) and by number of sessions included in training (1-5). These are plotted in Figure 5A. Multi-day training, reduced the initial decrease in performance (within ~5 days of training), improving performance over time, but did not seem to impact the longer term decay, and slightly decreases day 0 performance. A double-exponential fit (Fig. 5A) to the single-session and 5-session conditions over all evaluation sessions (within 180 days of day 0) show steeper decays in the single-session case, confirming our observations: the initial decay decreases with more training sessions, but the long-term decay is not significantly affected.

A complementary approach to multi-session pretraining is continual fine-tuning small amounts of (labelled) incoming data, gradually adapting the decoder to shifting neural signals while removing the need to collect entire training sessions every day. To simulate this, we trained an LSTM on a seed day,

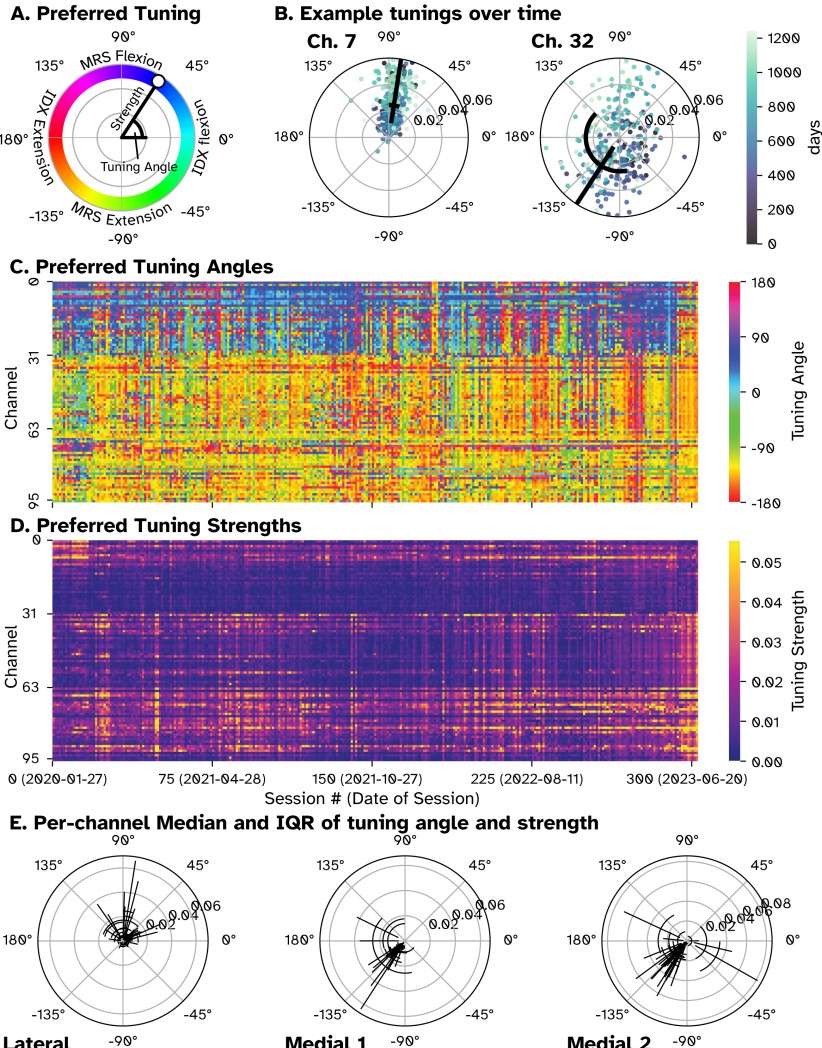

Figure 3: **Single-channel preferred tuning over time.** We investigated the relationship between the SBP of individual neural channels to behavior over the entire dataset by measuring the preferred tuning vectors of each channel for every session. In general, channels tended to stay tuned to the same direction throughout the dataset, with varying tuning strengths. **A.** Toy example illustrating how to interpret preferred tuning for a channel. Preferred tuning is a vector, measured on SBP, whose angle denotes the 'preferred' movement direction, and whose magnitude indicates the 'strength' of tuning towards this angle. **B.** Here we show the preferred tuning of two channels throughout the dataset. Each point shows the tuning of a single day, colored to indicate its temporal position in the dataset. The black crosses are centered at the median and circular median of the tuning strength and angle, respectively, while the size of the line and arc indicate the IQR (inter-quartile range) and circular IQR of the tuning strength and angle, respectively. **C.** Preferred tuning angles for all channels on all days. The preferred angle for each channel on each day in the dataset is colored according to the cyclical color bar and presented as a heatmap **D.** Preferred tuning strengths for all channels on all days. The preferred tuning strength for each channel on each day is colored according to the colorbar and presented as a heatmap. Any breaks between sessions are removed for visual clarity **E.** Per-channel tuning ranges across the dataset, grouped by bank. Channels were recorded in 32-channel banks, one from one array, and two from another. Here, we capture the range of tunings measured for each channel by plotting the circular median/median of the preferred angle/strength for each channel, and the circular IQR/IQR of each channel as error bars. We separated each bank into separate plots.

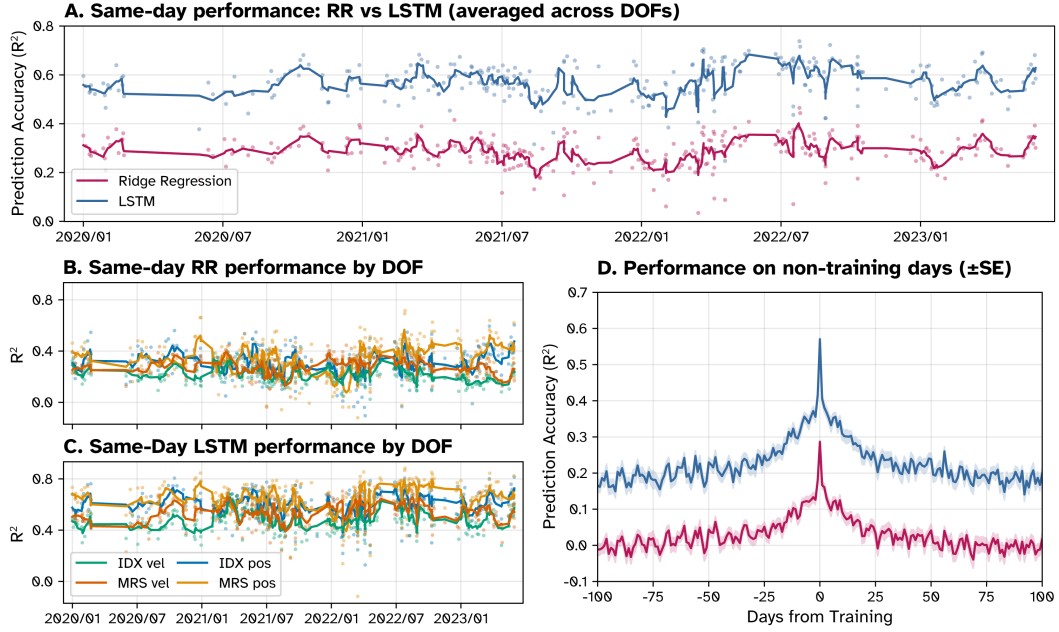

Figure 4: **Baseline decoding performance over the whole dataset.** Each session was split into a 300-trial training dataset and 75-trial hold-out, used to train two decoders, ridge regression (RR) and a long-short term RNN (LSTM). Prediction accuracy was measured using $R^2$. **A.** $R^2$ of each decoder tested on the same-day hold-out set. Points show per day $R^2$ averaged across DOF and the line shows the 5-day rolling average. **B-C.** Same-day performance separated by DOF for RR (B) and LSTM (C). **D.** Decoders were evaluated on hold-out sets within 100 days of training. Each line shows the average $R^2$ across all evaluations K days from decoder training, separated by decoder. Day 0 $R^2$ for each decoder is the average of the points in A. $\pm$ Standard errors are shown as shaded regions. Neither RR nor LSTM generalized well over time.

and then, for up to 200 days following training, we fine-tuned the LSTM on each consecutive session using 30 s to 5 minutes of training data. We repeated this for 20 seed days approximately evenly spaced across the dataset. We then plotted the average $R^2$ across the 20 seed days for each relative day from initial training, as shown in 5B. Fine-tuning with as little as 30 s of data helped reduce both the short and long-term decays in accuracy. Note that since fine-tuning was performed continually (e.g. a decoder fine-tuned on day 200 had also been fed data from all prior available sessions), performance generally improved gradually over time. In fact, fine-tuning on 120s or 300s led to later decoders outperforming the baseline day 0 decode. These results suggest that daily finetuning with relatively small amounts of labeled data can help with recovering performance.

## 5 Limitations, Discussion, and Conclusions

To our knowledge, LINK is one of the largest and longest spanning public releases of behaviorally-relevant intracortical data to date. The dataset contains two standard neural features (for iBMI decoding and neural population analysis), with two different tasks and details about each of the 96 channels. This data is provided for hundreds of trials per day, on hundreds of days, spanning almost 4 years. We believe the LINK dataset fills a critical gap and will enable new and interesting research in BMI decoding and neuroscience. However, the dataset has multiple limitations. First, the dataset only contains one subject, which may bias data. Secondly, the dataset contains multiple extended time gaps, some of which span months, mostly due to clusters of days failing to meet quality criteria (see Supplementary Materials). Additionally, due to storage limitations, raw voltage signals were not recorded for the majority of the sessions included in the dataset, limiting its use for developing new neural features or decoding methods which use raw data. Finally, due to the trial-based, constrained nature of the task, the actual task itself is not as behaviorally rich nor broadly labeled as might be desired for the development of foundational models for neuroscience [70, 71].

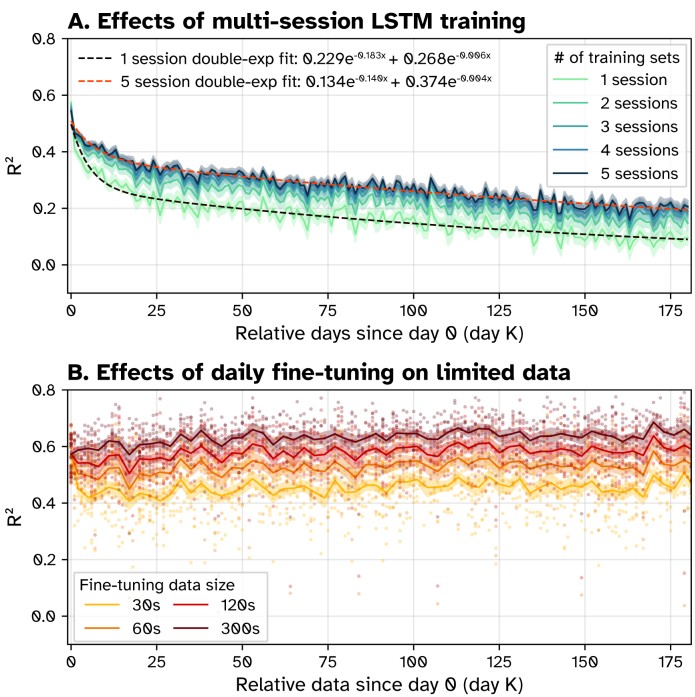

Figure 5: **Modified training paradigms for recalibration.** Here we test the ability of two training paradigms to improve the performance of LSTM decoders over time. **A.** On 283 days, 5 LSTM decoders were trained, using from 1-5 past sessions of data (current day inclusive). The plot shows the average prediction $R^2$ of the decoders tested on future sessions, across all models, grouped by the number of sessions included in the training data. **B.** We also tested performance when the LSTM model was fine-tuned on small amounts of future data every day, up to 200 days.

Figure 2 demonstrated that across all channels, neural activity generally decreased over time. This has been previously observed in [21], and is expected, as the array is gradually encapsulated by scar tissue over time [27]. However, this did not correspond to decreases in decoding performance throughout the dataset, suggesting that sufficient behavioral information remained for high-accuracy decoding. This is further supported by observations where, when projected into a low-dimensional neural space, neural trajectories for different directions remained separable throughout. More advanced and neuro-focused methods for latent feature extraction/dimensionality reduction (e.g. LFADS [48, 39], jPCA [47], dPCA [46], Isomap [72, 73]) may provide further insight into the evolution of population-level activity over time. In future datasets, it would be interesting to see if the gradual decrease in neural activity has an earlier observable impact on decoding accuracy for more complex tasks.

In Figure 3, we observed that many channels maintained a relatively stable preferred tuning direction over time (while tuning strength varied). Given that SBP and TCR capture the spiking activity of individual neurons close to the electrode [63], this stability in tuning direction could suggest that many electrodes are recording from the same neurons. We also observed that channels located on the same array tended to exhibit similar tuning angles, as could be expected from a somatotopic organization [74]. A potentially novel observation from the preferred tuning directions is the apparent separation of MRS flexion and extension tunings by array, but this data comes from only a single subject and task, and such a claim would require extensive investigation across several subjects.

In general, single-day decoding performance was fairly consistent throughout the lifespan of the array. While the LSTM outperformed RR on every day, we noted that there were baseline shifts in decoding performance that impacted both decoders, suggesting there are underlying shifts in the behaviorally-relevant information available in the neural data. When testing decoders on past and future data, we noted that both RR and LSTM exhibited sharp decreases in performance over the short-term, agreeing with previous observations [32, 26]. However, this decreases stabilizes on longer timescales, suggesting that there may be a stable subspace of neural activity that is at least partially behaviorally-relevant and nonlinearly related to behavior (hence the increased performance of the

LSTM decoder). This may also be supported by the shape of the decay being agnostic to the decoder used and the relatively consistent tuning directions of single channels.

## Acknowledgments and Disclosure of Funding

First, we would like to acknowledge Eric Kennedy, whose help has been indispensable. We would also like to acknowledge Monkey N for his years of contribution and hard work.

**Funding**
This work was supported by:
NSF Grant 2223822
NSF Grant 1926576
The Kahn, Dan D. and Betty, Foundation (AWD011321)
LHC was supported by Agencia Nacional de Investigacion y Desarrollo (ANID) of Chile
RP and AD are supported by a CASI award (1021865.01) from the Burroughs Wellcome Fund.

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
