# Supplementary Materials and Appendices for: Long-term Intracortical Neural activity and Kinematics (LINK): An intracortical neural dataset for chronic brain-machine interfaces, neuroscience, and machine learning

## A  Experimental Setup Details

### A.1  Behavioral Task

A male nonhuman primate (NHP, Macaca mulatta), Monkey N (age 7 at the beginning of the dataset, age 11 at the end), was trained to perform a trial-based, two degree-of-freedom (DOF) dexterous finger movement task, shown in Figure 1. During all sessions, Monkey N sat in a primate chair (Crist Instruments, Hagerstown, MA) in a shielded chamber, with his arms fixed at his sides and flexed 90 degrees at the elbow, resting on a table. The left hand was positioned securely in a manipulandum, which used bend sensors (FS-L-0073-103-ST, Spectra Symbol, Salt Lake City, UT) to measure the flexion of two finger groups, index (IDX) and middle-ring-small (MRS). At the beginning of each experimental session (and as needed throughout a session), these flexion sensors were calibrated such that a reading of 1 indicated full flexion of a finger group and a reading of 0 indicated full extension. These readings were used to update the positions of the corresponding finger groups of a virtual hand presented on a screen in front of Monkey N. Bend sensor values were sampled at 1000 Hz. Updates to the virtual hand were limited to the refresh rate of the monitor (120 Hz).

The task itself involved trial-based target acquisitions. At the beginning of each trial, two color-coded spherical targets, one for each DOF, were placed on the screen, covering 15% of the full arc of motion (see Figure 1A). Monkey N then acquired the targets by moving his fingers to the correct positions and holding his position for 750 ms. Upon successful completion of a trial, Monkey N received a juice reward. If targets were not acquired within a trial timeout (typically 5 or 10 seconds), the trial was considered a failure, the monkey was provided with no juice reward, and the next set of targets was presented. Trials were performed in continuous blocks, known as 'sessions'. We used Fitt's law to measure per-trial throughput (bits per second) as a metric of real-time perofrmance, as described in [1]. In S2, we show the average throughput across all trials for each day, and noted that Monkey N's performance appears to gradually improve throughout the course of the dataset.

Depending on the session, targets were presented to Monkey N in a variety of patterns, referred to as 'target styles'. Two target styles are included in the LINK dataset, representing two common tasks in motor BMI research, 'center-out' and 'random'. Both are shown in Figure 1D and are described below:

**Center-out**  The center-out (CO) pattern mimics the center-out-and-back pattern described in previous studies [1]. In this pattern, every other trial is a return to a 'rest' position of [0.5, 0.5]. On non-rest trials, or 'reaches', were randomly selected from a discrete library of 8 'directions' of movement (4 single DOF movements, 2 multi-DOF movements in the same direction, and 2 split movements) and three magnitudes of movement (20%, 30%, or 40% of the movement range), shown

39th Conference on Neural Information Processing Systems (NeurIPS 2025) Track on Datasets and Benchmarks.

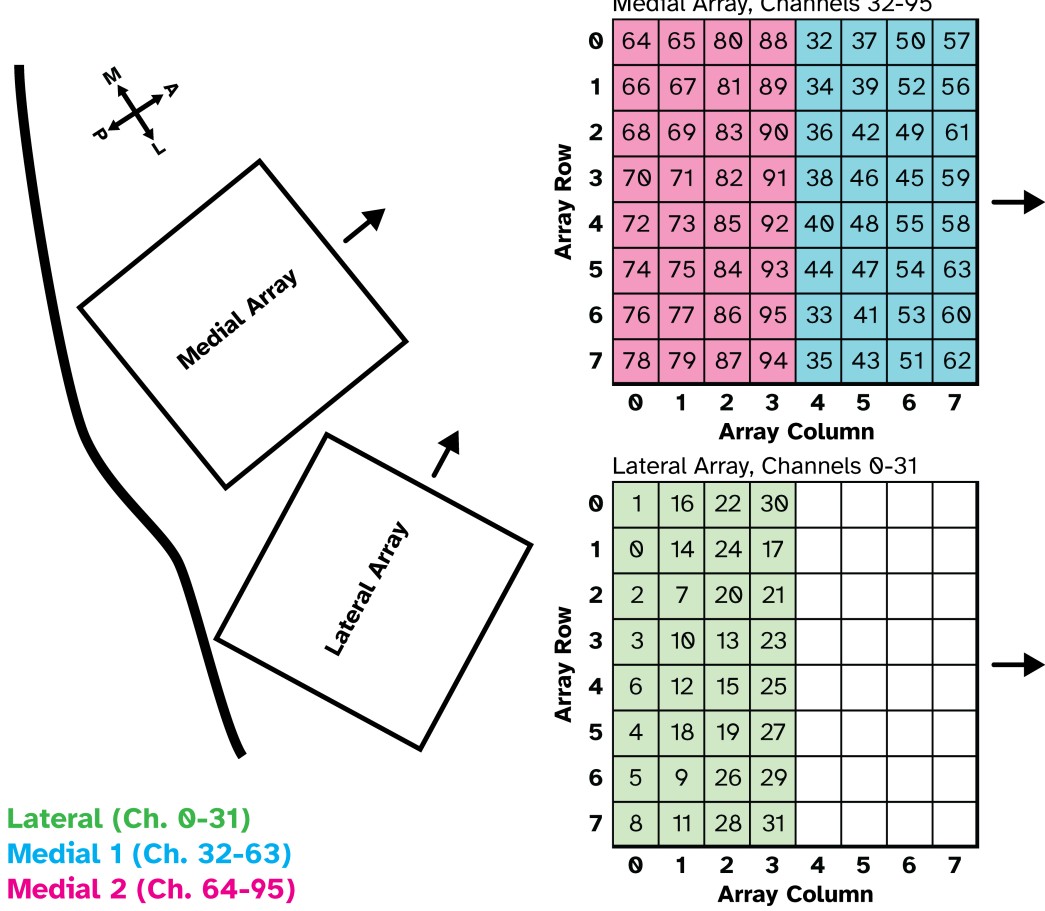

Figure S1: **Array map.** Relative spatial position of each channel in each array, numbered 0-95. On the left, a visual diagram shows the relative positions of the 2 $8 \times 8$ Utah MEAs. The black arrows represent the direction of the wire bundle leading to the CerePort pedestal and recording equipment. On the right, we see a map showing the relative spatial positions of each channel included in the LINK dataset. For example, channel 47 is on the medial array, in row 5, column 5. The arrows represent the relative wire bundle direction for orienting to the diagram on the left.

in Figure 1C. Since Monkey N had difficulty performing split movements large than 50% of the movement arc, the split targets did not include the 40% movement magnitude.

**Random** In the random (RD) target style, new targets were selected at pseudo-random positions for each trial. To pick movement targets, first a random separation between the finger groups was sampled ($f_{sep}$, 0 to 0.5 from uniform random distribution) and then a midpoint was sampled (uniformly between $f_{sep}/2$ and $1 - f_{sep}/2$).

## A.2 Implants

Monkey N was implanted with two 64-channel Utah microelectrode arrays (MEA, Blackrock Neurotech, Salt Lake City, UT) in the right precentral gyrus, using the arcuate sulcus as an anatomical landmark for the hand area, as previously described in [2, 3]. An (unused) 96-channel Utah array as immplanted in the somatosensory cortex, as shown in Figure 1. Electrode shanks were 1.5mm long, targeting layer V of the motor cortex. The implant locations can be seen in Figure 1B. While 128 channels were available for recording on the motor arrays, only 96 channels were recorded due to hardware limitations. These 96 channels were recorded in three banks of 32 channels each, indicated in Figure 1C. The recordings included in the dataset begin on day 349 post-implant, up to 1591

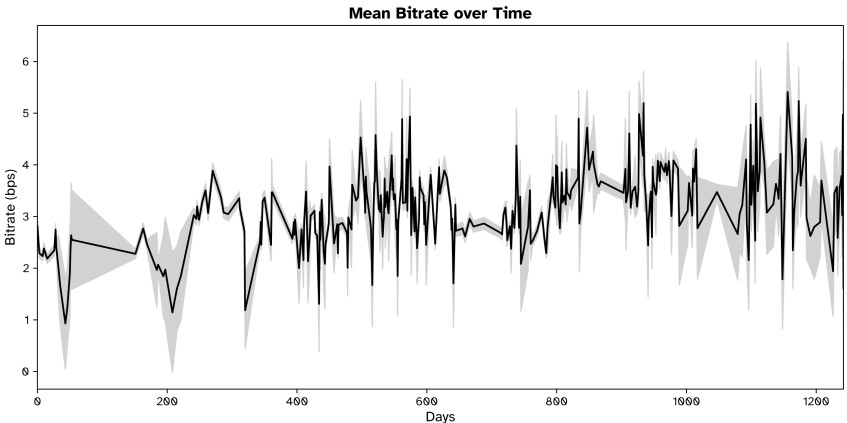

Figure S2: **Average trial throughput per day.** Throughput (measured in bits per second, Fitt's Law) was calculated per trial. Here, the black line shows the average throughput across all trials in a single day (note that 9 days include 2 sessions). The shaded area shows the standard error about the mean.

days post-implant. Throughout the lifespan of the array, we recorded impedance measurements to assess array health approximately weekly. Each electrode is labeled with the impedance measurement closest in time to the date of the session. If no impedances were recorded within three weeks of a session, no impedances are reported for that session. The protocols in this study were approved by the Institutional Animal Care and Use Committee at the University of Michigan.

### A.3    Signal Processing and Feature Extraction

During each session, we recorded 96 channels of data from the implanted MEA using a Cerebus Neural Signal Processor (NSP, Blackrock Neurotech, Salt Lake City, UT). Raw neural data was sampled by the NSP at 30 kHz, but was processed, synchronized, and downsampled in real-time during experimental sessions into two neural features, threshold crossing rate and spiking-band power (SBP) [4], as shown in Figure 1B. The two features and how we processed them are described below.

**Threshold Crossings (TCR, Spike Counts)**    Threshold crossing rate (TCR) is an analog for the spiking rate of the neurons closest to each electrode's recording site. TCR was obtained on each day by configuring the NSP to extract voltage snippets that cross a threshold of -4.5 times the root-mean-square of each channel, calculated at the beginning of each experimental day using the Central Software Suite and Cerebus NSP (Blackrock Neurotech, Salt Lake City, UT). Thresholds were calculated for each channel at the beginning of a session. Threshold crossings were sent to a computer running xPC Target version 2012b (Mathworks, Natick, MA, USA), which recorded the channel and arrival time of each spike in 1 ms bins.

**Spiking-band Power (SBP)**    Spiking-band power (SBP) is a low-power feature [4] which contains the majority of single-unit spiking activity of the neurons closest to each electrode. It was obtained by configuring the NSP to band-pass filter the raw signals on each channel to 300-1,000 Hz and sampling at 2 kHz. The filtered signal was then sent to the same computer running xPC Target, which took the magnitude of each incoming sample, and summed all samples received within 1 ms bins. This corresponds to an efficient estimate of the signal power within the band of interest.

**Binning**    The 1 ms resolution features (TCR, SBP, finger flexion) were further binned into non-overlapping 20 ms bins. To do this, the TCR counts were further summed into 20 ms bins (from the 1 ms bins already compiled); average SBP was calculated by summing the SBP feature in each 1 ms bin and dividing by the total number of 2 kHz samples recorded in that 20 ms window; and finger flexion from all 1 ms intervals in the 20 ms bin was averaged together. Additionally, the finger flexion velocity was calculated by taking the difference of the 20-ms binned finger flexions. The time vector is extracted by tracking the number of 1 ms intervals since the experiment started. These 1 ms intervals are guaranteed by the real-time xPC computer that controls experiments.

# B  Dataset selection and format details

## B.1  Data Curation

Data was curated using a three-stage process. Each stage is described below. The second and third stages were performed iteratively until settling on the final dataset.

**Initial Candidate Selection**   As an initial vetting of experimental days, we reviewed all experimental notes from Monkey N from January 2020, the first month Monkey N was consistently proficient at performing the task, to February 2024, ~six months after neural activity and real-time decoder performance significantly declined. We identified candidate days and sessions within days that appeared to contain the correct target style, correct hold times, and had a sufficient number of trials (~400). We excluded days if the experimental notes reported obvious issues with the signal (e.g., no neural data present) or were running different experimental setups (e.g., an acute EMG stimulation experiment), or other problems like monkey behavioral issues or crashes to our data logging setup. Note that this was not a systematic review of each of the note files, just an initial filter of days denoted as unusable and a selection of candidate runs. After this first pass, we were left with 453 days spanning from January 2020 to February 2024.

**Automated Session Filtering and Preprocessing**   After being selected for further review, each session was loaded and filtered to remove unwanted trials. The following exclusion criteria were used to filter trials:

- First five trials, removed to prevent adaptation or 'warm-up' bias
- Any trials not adhering to the majority target pattern, and any trials not falling under the CO or RD target styles.
- Closed-loop brain-machine interface trials
- Trials with a hold time $\neq 750ms$
- Unsuccessful trials (trials exceeding the trial time out)

Unsuccessful trials were removed as they introduce confounding variables into the neural data, e.g. one trial may be unsuccessful because Monkey N had trouble reaching the targets, while another may have been unsuccessful because he removed his hand from the manipulandum entirely. He could still be generating brain activity, but it would be irrelevant to the behavioral labeling.

After filtering, we selected the first 375 trials of the remaining session. If a session had less than 375 trials, it was excluded in its entirety. If multiple sessions on the same day with the same target style had greater than or equal to 375 trials, the one with the largest overall trial count was included. If two runs had 375 trials and different target syles (i.e., one CO and one RD), both were kept. After this pruning process was complete, 416 days containing sessions remained. Once the trials for a particular session had been selected, we ran our feature extraction and binning process, explained above. Given a trial of $N$ total bins, the SBP and TCR neural features are vectors of dimensions $N \times 96$, time is $N \times 1$, and behavior is $N \times 4$–the positions and velocities for each of two DOFs. We also extracted relevant trial metadata, such as trial starts and trial lengths.

## B.2  Manual Data Review

After filtering trials, pruning 'bad' sessions, and preprocessing the dataset, all timeseries data (SBP, TCR, behavior) were inspected several times by three human reviewers. Reviewers inspected the entire dataset 10s at a time, flagging any sessions exhibiting hardware or behavioral confounds as 'bad'. Sessions were marked as 'bad' if channels showed excessive noise (this was typically cross-checked with experimental notes), if one or more channels was disconnected, if baseline shifts in resting SBP and noise were seen in all channels, decoder resets or misfires drove bursts of artifact signals, prolonged drop-outs (>1s), or large transient jumps suggestive of connector instability or excessive movement, or monkey disengaged behavior (e.g. sustained durations of no movement). At this stage, we also identified more experimental setups that introduced noise or otherwise prevented proper data recording (e.g., simulated asynchronous data replacing TCR, or turning on common average referencing halfway through a session). In most cases, we corroborated sessions with

ambiguous or noisy signals using experimental notes, checking for documented issues such as closed-loop decoder failure or irregular behavior. Additionally, several sessions from October and November 2020 were excluded due to their previous inclusion as hold-out datasets in the FALCON benchmark [5]. Any sessions recorded within one calendar day of these hold-outs were also excluded. After the manual data review, we were left with 312 sessions on 303 days, spanning 1242 days.

## B.3   Data Format

The LINK dataset is hosted on the DANDI archive at `https://dandiarchive.org/dandiset/001201`. Each session is stored as a separate file and follows the Neurodata Without Borders (NWB) standard. The session files are structured according to the BIDS (Brain Imaging Data Structure) format. NWB files are HDF5 files with a standardized hierarchy for organizing neural data and relevant experimental data. Here we document the structure of each session file. At the root level, we have a session description and identifier, and the following groups:

- `session_start_time`
- `timestamps_reference_time`
- `file_create_date`
- `experimenter`
- `analysis`
- `keywords`
- `electrodes`
- `electrode_groups`
- `devices`
- `intervals`
- `subject`
- `trials`

First, we will describe the least relevant groups: specific session start times are not included, so `session_start_times` and `timestamp_reference_time` are set to midnight of the date the session was recorded. `file_create_date`, `experimenter`, and `keywords` are self-explanatory. The `subject` group contains Monkey N's sex, date of birth, and species. `devices` contains 2 entries, defining the two $8 \times 8$ arrays we recorded from. These are referenced by `electrode_groups`, which contain similar information. The `intervals` group can contain other time intervals, but here it only references the `trials` group, which is described below. The following are descriptions of the groups with more data:

**Electrodes**   The `electrodes` group contains a table with one row per channel, for a total of 96, as indicated by the `id` column. The corresponding group and array for each is indicated by the `group_name` and `array_name` columns (these contain the same information but comply with the standard). The `bank` column indicates which 32-channel bank (A-C) was used to record the data, the `pin` indicates which bin on the bank, and the `row` and `col` columns indicate the relative position of each channel on its respective array, as shown in Figure S1. Impedance measurements as described above are included in the `imp` column.

**Trials**   The `trials` group contains a table with one trial per row and the following columns: `start_time` (in sec), `stop_time` (in sec), `trial_number`, `trial_count` (the length of each trial in 20 ms bins), `run_id` (not relevant, indicates what session this was within a day), `index_target_position` (from 0 to 1), `mrs_target_position`, the `target_style` used to generate the targets, the `trial_timeout` (time to failure, ms), and `timeseries`. The `timeseries` column contains a list of references to all the `timeseries` objects relevant to the trial (which in this case is all of them). These `timeseries` are described below.

**Analysis**   All time series in this dataset are included in the analysis group of each session. Each `timeseries` object contains a resolution, description, conversion, unit, and two HDF5 datasets, one with the actual data, and one with the corresponding timestamps. The `timeseries` objects for the two nearal features also contain a reference to the electrodes table. There are six timeseries: `SpikingBandPower`, `ThresholdCrossings`, `index_position`, `index_velocity`, `mrs_position`, and `mrs_velocity`. Each contains the full timeseries (not separated by trial).

To facilitate getting started with the LINK dataset, and code at `https://github.com/chesteklab/LINK_dataset` for converting these NWB files into dictionaries containing only trial and timeseries information. In this case, days with multiple sessions are saved as a single file, and all days are saved as .pkl files. A description of these can be found in the readme. The dataset and code can also be accessed through `https://chesteklab.github.io/LINK_dataset/`.

## C    Analysis Methods

### C.1    Decoders

New decoders were trained on every session, using the first 300 trials as training data and the final 75 trials as a held-out test set. Decoding analyses were performed with Python 3.10. The input neural features for both decoders were normalized to zero mean and unit standard deviation using the training data on each day. Both decoders predicted the position and velocity of the two finger groups (four total outputs) at each time step. Decoders were trained on a computer containing an AMD Ryzen Threadripper PRO 5965WX, 128gb RAN, and an NVIDIA RTX 4090. Individual networks trained in <30s, it took ~1 hour to train all the decoders and ~4 hours to run the evals on each type of decoder (9 hours total).

**Ridge Regression**    As a simple linear decoder, we trained ridge regression models using the `sklearn.linear_model.Ridge` class, using alpha = 0.1. The decoders used 8 bins of time history for each channel, for a total of 768 input features.

**Long Short-Term Memory Recurrent Neural Network**    As an example of a neural network decoder, we trained LSTM models using the torch.nn.LSTM class with PyTorch 2.6.0. The decoder took in 96 features as input, used a hidden size of 300 with 1 layer, and used a fully connected layer to project the hidden states to the four outputs. The decoder was trained with a sequence length of 20 bins, and kinematics were predicted at the final timestep. Decoders were trained using the Adam optimizer with batch size 64, trained for 2000 iterations, and used a linearly decaying learning rate of 2e-4 to 1e-5. During training, a small amount of Gaussian noise was added to the neural activity, which has been shown to improve performance, using an overall noise standard deviation of 0.1 and a bias noise standard deviation of 0.2. More details on training can be found in Costello et al. 2023 [6].

**Decoder Evaluation**    Decoders were evaluated both on the same day as their training data and across all other days in the dataset, using the 75 held-out trials on each day. For the analyses in Figure 5, the channel normalization parameters were held constant, using the parameters found on the training day. Decoders were evaluated by comparing the predicted kinematics at each time step to the true kinematics for each of the four kinematic outputs. All decoders were evaluated by measuring the coefficient of determination ($R^2$) as a measure of prediction accuracy. $R^2$ was chosen over MSE because MSE would need to be separate for velocity and position as they have different units and often different orders of magnitude. Additionally, these units are not intuitive (e.g. velocity is proportion of flexion/20 ms bins). $R^2$, on the other hand, provides a more interpretable metric on a 0-1 (and sometimes negative) scale. As a reference point for interpreting model performance of the RR and LSTM models, on one session, we independently shuffled the neural and behavioral data of the holdout set, and predicted behavior using RR and LSTM models trained on that day. We then measured the average $R^2$ of these predictions with the shuffled behavior data. RR had an average $R^2$ of -0.166, and LSTM had an average $R^2$ of -0.156.

### C.2    Preferred Tuning Direction

To measure the relationship between single-channel activity and kinematics, we calculated a metric we call 'preferred tuning direction'. On each day, for each channel, we fit a linear regression between the normalized SBP for a whole session and the 2-DOF (either position or velocity). This produced two coefficients, for index and MRS, which we stack into a vector. We then measure the magnitude of this vector (referred to as the 'tuning depth' or 'tuning strength') and the angle of this vector ('tuning angle'). The angle indicates the direction of movement which causes the most firing, and the magnitude indicates the strength of this relationship. For example, a tuning angle of 0 degrees indicates that the channel has higher activity during index flexion movements.

Since there is a potential lag between a neuron firing and a behavior being performed, we lagged behavior relative to neural data relative (behavior lag, i.e. testing $neural_t$ with $behavior_{t+\Delta t}$), calculated tuning at each lag, and took the tuning which produced the greatest magnitude (or L2 norm). we tested up to 10bins of lag (200ms). This lag optimization was performed on a per-channel basis on every dataset. This was similar to the approach taken by [7].

Tunings were visualized using various approaches depending on the focus. The magnitudes were limited to 0 through 0.06 (rounded 0.01 and 0.99 quantile of magnitudes over all channel and time). The angles ranged from -180°to 180°. When investigating single channel tuning over time, a polar plot was used and multiple data points over time were plotted with magnitudes and angles. Different colours were assigned to each datapoint according to the day from day 0. When investigating multiple channel tuning over time, heatmaps were used by having dates on x-axis and channels on y-axis. The colours were used to represent the values of each part of tuning, including magnitude and angles. The dates without any data were set to grey. When investigating the spread of tunings, polar plots are used with the mean and interquartile range of magnitudes as well as the circular median and circular interquartile range of the angles.

Circular medians were calculated by minimizing equation 1 from [8]. It can be further inferred from equation 1 that the median can be found by minimizing the shortest angular distance between all angles to the target angle. This method was implemented in a brute-force way.

$$d(\theta) = \pi - \frac{1}{n} \sum_{i=1}^{n} |\pi - |\theta_i - \theta||$$ (1)

The circular interquartile (cIQR) range is calculated by calculating the difference between circular quartiles. cIQR is calculated by shifting the circular median to 0, calculating quartiles for angles, and shifting the quartiles back. We adapted our Python code implementation from the Circular package in R [9].

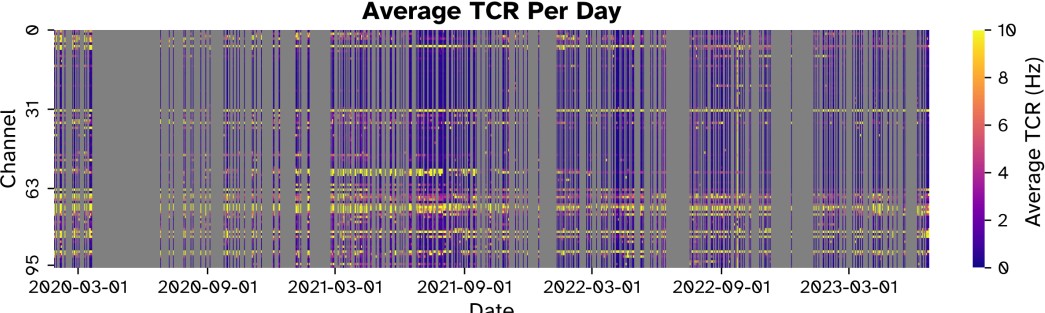

Figure S3: Heatmap showing the average TCR, per channel, on each day included in the dataset. Each cell on the heatmap is colored according to the colorbar on the left. Days not included in the dataset are colored gray.

We noted in Figure 3 that the preferred tuning directions exhibited some similarities depending on the physical location of electrodes on the cortical surface. Furthermore, the channels with high tuning strengths correlated with channels deemed 'active' by their TCR activity, as seen in S3. It may be interesting to look at average SBP changes within these groups, similarly to what was shown in Figure 2A. Grouping by 32 channel 'banks' (as shown in S1) gives us the plots in Figure S4. The slopes of the best-fit lines for the Lateral, Medial 1, and Medial 2 groups were $-0.58 \times 10^{-3}$, $-1.34 \times 10^{-3}$, and $-1.40 \times 10^{-3}$ respectively. This is compared to $-1.11 \times 10^{-3}$ for the overall SBP. This is largely because the average SBP is generally higher on the medial arrays, which makes sense as they are in the motor cortex. Grouping channels within each session by active channels (average TCR < 1Hz) vs. inactive channels gives us the plots in Figure S5. The slopes of the best-fit lines here were $-1.29 \times 10^{-3}$, and $0.54 \times 10^{-3}$. As might be expected, the slope for the inactive channels was near 0, as the channels were inactive and thus not likely to have high SBP activity in the first place. The slope for active channels was higher than the overall slope of $-1.11 \times 10^{-3}$, as expected.

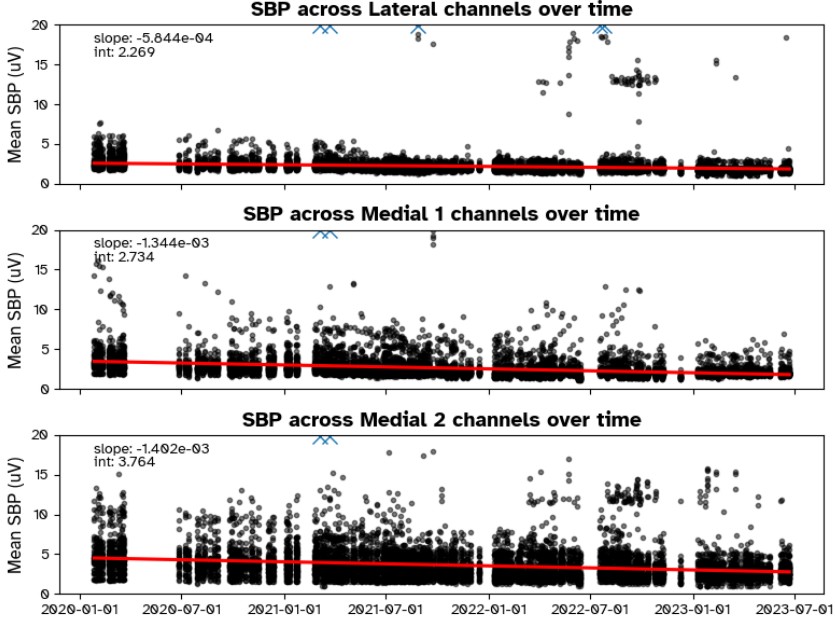

Figure S4: **Average SBP across brain regions** Channels were deemed inactive or active based on threshold crossings, and the mean SBP for each electrode bank was calculated for each day.

### C.3 Neural Population Analyses

**Changes to neural population over time** In order to visualize changes to the neural data over time at a population level (Figure 2D), we z-scored SBP data, per channel, concatenated across the whole LINK dataset. We then fit a PCA transform using the `scikit-learn` library to the full-dataset SBP matrix. Data was transformed and reduced to 3 principal components, and then grouped by day and by quarter (of a year). The centroids and standard deviations of the transformed data in the low-dimensional subspace were then calculated per day and quarter.

**PCA trajectories grouped by movement, across years** To visualize how behaviorally-relevant data in the neural population changed over time, as shown in Figure 2E, we performed a similar process as described above. However, we z-scored SBP per-channel and fit a PCA transform per-day, rather than across the entire dataset. We then grouped trials by year and time-aligned and trimmed all trials as described below. Then we further grouped these trimmed trials by movement direction, and described below, and calculated the average trajectory of the top 3 PCs of the PCA-transformed SBP data. The viewing angle of the plots was chosen to maximize the visual separability of the trajectories.

**Grouping Trials by Direction** To discretize the movement space, all trials were labeled by the corresponding 'movement direction' required to complete the trial. Movement direction was defined as flexion or extension of one or both DOF (IDX and MRS), for a total of 8 possible directions. To do this, each DOF was visualized as an axis in a two-dimensional plane, and we measured the angle of the displacement vector between a trial's starting position (the previous trial's target positions) and a trial's target positions. We then assigned a trial to a movement direction by splitting these angles into eight $45°$ sectors. This approach ensures that movements driven primarily by a single finger group (e.g. a movement from $[IDX, MRS]$ position $[0.5, 0.5]$ to $[0.55, 0.8]$) are assigned accordingly, even if both DOF need to move, as is the case in most random target style trials. The classification for all trials across the dataset can be seen in Figure S6. Note that the max separation between IDX and MRS targets in a trial is constrained to 0.5, which leads to the skew observed in Figure S6.

**Alignment by movement onset** To account for noise in low-dimensional neural trajectories due to misalignment in neural activity, we time-aligned trials within each movement direction group using the maximum jerk before movement onset. We determined movement onset as the first point

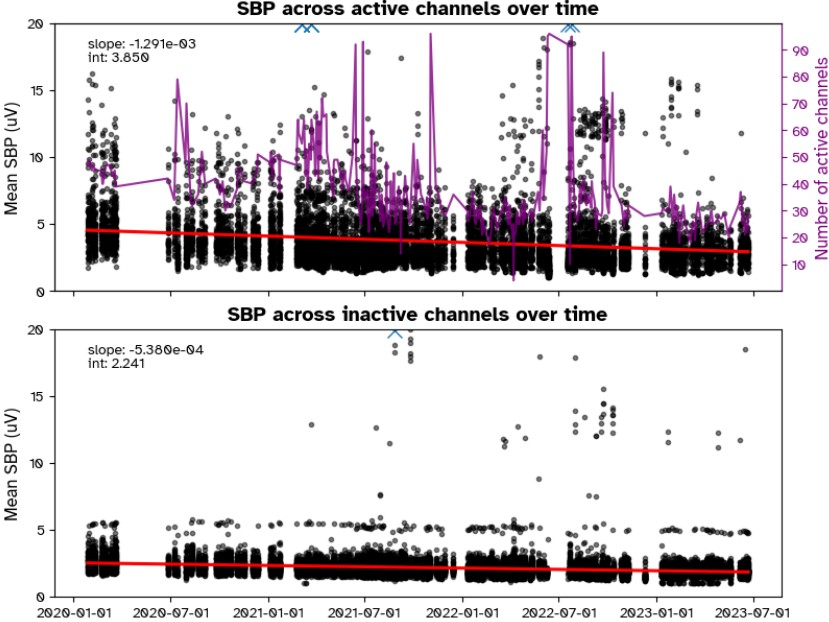

Figure S5: **Average SBP across active and inactive channels.** Channels were deemed inactive or active based on threshold crossings, and the mean SBP within each day was was calculated for both activity groups. The purple line shows how many channels were deemed active on each day.

where finger velocity exceeds one standard deviation from the mean velocity of the first 10% of the trial's velocities. The finger velocity used in this calculation depended on the type of trial (see the grouping paragraph above). If the movement direction was MRS-dominant, MRS velocity was used. Otherwise (including for dual finger movements) IDX velocity was used. Jerk was computed by taking the second discrete difference of velocity. Due to the varying length of trials, trials were trimmed post-alignment to 740 ms (37 bins) starting from the point of alignment.