# OpenReview forum: "Long-term Intracortical Neural activity and Kinematics (LINK): An intracortical neural dataset for chronic brain-machine interfaces, neuroscience, and machine learning"
_NeurIPS.cc/2025/Datasets_and_Benchmarks_Track — NeurIPS 2025 Datasets and Benchmarks Track poster_

### Official Review · Reviewer_fb9y · 2025-07-02

**Rating:** 4
**Confidence:** 4

**Summary:**

This paper publishes an open-source dataset named LINK, which contains intracranial neural activity and kinematic data from a non-human primate performing fine finger movement tasks over 1,242 days (approximately 3.5 years). The dataset aims to provide a standardized evaluation platform for addressing the problem of long-term instability of neural signals in brain-machine interfaces (BMI).

**Additional Feedback:**

Would it be possible to provide the raw voltage signals? This would be beneficial for research on neural decoding.

In the Analyses section, in addition to showing the overall decrease in signal strength (SBP), a more detailed analysis could be added to investigate whether this signal decay differs across different brain regions (e.g., the medial vs. lateral arrays) or among different types of neurons (e.g., those with high vs. low tuning strength).

**Dataset Code Accessibility:**

Partly

**Dataset Code Comments:**

The dataset has been open-sourced on public websites and provides data visualization codes and data preprocessing codes.

**Ethical Considerations:**

No, there are no or only very minor ethics concerns

**Limitations Weaknesses:**

1. Since there was only one subject, the data may be biased.
2. Although the paper preliminarily tested some models, it does not include learning algorithms specifically designed for long-term neural signals, such as lifelong learning or continuous learning.
3. It provides preprocessed neural features rather than raw voltage signals.

**Strengths Contributions:**

1. This paper is well-written and easy to understand
2. The problem of long-term instability of neural signals evaluated in the paper is a key issue in the field.
3. Although the signal intensity of individual channels varies, the activity patterns of neural populations (trajectories in low-dimensional space) maintain long-term separability across different movement directions, indicating that the neural "manifold" encoding motor information is relatively stable.

---

> ### Author Rebuttal · Authors · 2025-07-30
>
> Thank you for the helpful comments, which we address below:
>
> **Single subject**: We completely agree. We’ve modified the limitations section of the discussion to read:
>
> >“However, the dataset has multiple limitations. First, the dataset only contains one subject, which may bias data. Secondly, the dataset contains multiple…”
>
> **Continuous learning**: Thank you for pointing this out! Reviewer TZPf also pointed this out, so we add some relevant text and analyses accordingly. Please see our response to their comment for more details and a summary of what’s been added, including pretraining across days and fine-tuning on future days as a continual learning paradigm.
>
> **Preprocessed neural features vs raw voltage traces**: Thank you for mentioning this. Unfortunately, it is not possible to supply all the raw data as it was not stored and often not used for decoding purposes. We have modified the relevant sentences in the limitations section of the discussion to read:
>
> >“…quality criteria (see Supplemental). Additionally, due to storage limitations, raw voltage signals were not recorded for the majority of the sessions included in the dataset, limiting its use for developing new neural features or decoding methods which use raw data. Finally, due to the trial-based..”
>
> **Signal decay across regions**: This is a great idea. We conducted a new analysis by grouping channels by region and by TCR activity (closely related to tuning strength), and will add a figure to the supplemental materials showing multiple plots similar to Figure 2A. Additionally, we will add the following paragraph to the Supplement:
>
> >“We noted in Figure 3 that the preferred tuning direction of channels seemed to be grouped by the physical location of their Utah array, and that the channels with high tuning strengths correlated with channels deemed ‘active’ by their TCR activity. It may be interesting then, to see how average SBP changes within these groups, similarly to what was shown in Figure 2A. Grouping by 32 channel ‘banks’ (1 of which was on the lateral array, 2 of which comprised the medial) gives us the plots in Figure SXA. The slopes of the best-fit lines for the Lateral, Medial 1, and Medial 2 groups were -0.58 x 10-3, -1.34 x 10-3, and -1.40 x 10-3 respectively. This is compared to -1.11 x 10-3 for the overall SBP. This is largely because the average SBP is generally higher on the medial arrays, which makes sense as they are in the motor cortex. Grouping channels within each session by active channels (average TCR < 1Hz) vs. inactive channels gives us the plots in Figure SXB. The slopes of the best-fit lines here were -1.29 x 10-3, and 0.54 x 10-3. As might be expected, the slope for the inactive channels was near 0, as the channels were inactive and thus not likely to have high SBP activity in the first place. The slope for active channels was higher than the overall slope of  -1.11 x 10-3, as expected. “

---

> > ### Comment · Area_Chair_NGEF · 2025-08-05
> >
> > Reviewer fb9y, please respond to the authors' rebuttal as soon as possible. The author–reviewer discussion period is nearing its end.

---

> > ### Comment · Reviewer_fb9y · 2025-08-06
> >
> > Dear authors：
> > I appreciate that you have acknowledged the limitations of the work, and I commend the new analyses conducted in response to the feedback, particularly concerning the continuous learning paradigm and the signal decay across different regions. These additions have certainly improved the manuscript.
> >
> > However, my primary concern regarding the dataset's broader utility and generalizability remains. While you have now explicitly stated the limitations in the discussion, the fundamental issues persist. The dataset is derived from a single subject, which significantly limits claims of generality. More critically, the lack of raw voltage traces for the majority of sessions is a major drawback. This prevents the dataset from being used to develop or validate novel feature extraction techniques or decoding algorithms that operate directly on raw neural signals, which is a key direction for the field.
> >
> > Therefore, while I recognize the diligent efforts made in the rebuttal, these core limitations of the dataset itself prevent me from raising my score at this time.
> >
> > Thank you again for your thorough engagement with the review process

---

### Official Review · Reviewer_8BwM · 2025-07-03

**Rating:** 5
**Confidence:** 4

**Summary:**

The authors present a long-term intracortical neural activity and kinematics dataset (LINK), which contains intracortical spiking activity and kinematic data from 312 sessions of a non-human primate performing a dexterous, 2 degree-offreedom finger movement task, spanning 1,242 days. They also demonstrate some longitudinal analyses of the dataset, showing the relationship between neural recordings and behavioral kinematics.

**Dataset Code Accessibility:**

Yes

**Dataset Code Comments:**

The dataset is well curated and easy to access. The related code is also opensourced and well documented.

**Ethical Considerations:**

No, there are no or only very minor ethics concerns

**Final Justification:**

I acknowledge the author’s helpful answers to my concerns which strengthens my confidence. I will keep my rating consistent.

**Limitations Weaknesses:**

1. The long-term stability of electrode recordings and their impact on signal acquisition/quality require further discussion.

2. The authors have applied PCA to obtain latent variables. However, there are more advanced methods for this task, like LFADS [1] and many others, which are worth trying.

3. A word 'trials' is redundant in this sentence and can be deleted.   "Sessions were grouped by year, and trials were labeled trials by the direction of movement required (as shown in Figure 2E)."

4. Some contents in the supplementary materials should be referenced in the main text to make things clearer. E.g., the method to calculate tuning preference, the meaning of IQR, and so on.

5. What's the exact meaning of the last sentence in Sec.5?   "Paired with the observation that many channels did not change much in their tuning angles, we think this could provide some foundation for developing alignment methods that aren’t too computationally expensive."

[1] Pandarinath, C., O’Shea, D.J., Collins, J. et al. Inferring single-trial neural population dynamics using sequential auto-encoders. Nat Methods 15, 805–815 (2018). https://doi.org/10.1038/s41592-018-0109-9

**Strengths Contributions:**

1. The dataset contains 312 sessions on 303 days, spanning ~3.5 years of a single monkey performing a trial-based dexterous finger movement task, which can serve as a testbed for longitudinal iBMIs, nonstationarity analysis in neuroscience, or developing new domain adaptation approaches for timeseries data more generally.

2. Each session of the dataset is contained in an HDF5 file compliant with the NWB standard, thus is well curated, and user-friendly.

3. The authors also present some initial analyses on the quality of neural signals, their relationships with behavior, and some initial iBMI decoding results.

4. In general, the paper is well-written and organized.

---

> ### Author Rebuttal · Authors · 2025-07-30
>
> Thank you for the helpful comments, which we address below:
>
> **Long-term stability**: We agree, and we’ve updated the relevant paragraph of the discussion to read:
>
> >“In Figure 3, we observed that many channels maintained a relatively stable preferred tuning direction over time (while tuning strength varied). Given that SBP and TCR capture the spiking activity of individual neurons close to the electrode [62], this stability in tuning direction could suggest that many electrodes are actually recording from the same neurons over the whole dataset. We also observed…”
>
> **PCA or LFADS**: We completely agree that these analyses would also be very appropriate. This paper is meant to be a starting point for applications of the LINK dataset, but we did indeed fail to mention other methods for projecting into a latent space. To remedy this, we’ve included the following sentence to the discussion:
>
> >“…for different directions remained separable throughout the dataset. In addition, more advanced and neural-focused methods for latent feature extraction or dimensionality reduction (e.g. LFADS [1,2], jPCA [3], dPCA [4], Isomap [5,6]) may provide further insight into the evolution of population-level activity over time. In future datasets, it would be interesting…”
>
> [1] Sussillo, D., Jozefowicz, R., Abbott, L. F., & Pandarinath, C. (2016). Lfads-latent factor analysis via dynamical systems. arXiv preprint arXiv:1608.06315.
>
> [2] Pandarinath, C., O’Shea, D. J., Collins, J., Jozefowicz, R., Stavisky, S. D., Kao, J. C., ... & Sussillo, D. (2018). Inferring single-trial neural population dynamics using sequential auto-encoders. Nature methods, 15(10), 805-815.
>
> [3] Churchland, M. M., Cunningham, J. P., Kaufman, M. T., Foster, J. D., Nuyujukian, P., Ryu, S. I., & Shenoy, K. V. (2012). Neural population dynamics during reaching. Nature, 487(7405), 51-56.
>
> [4] Kobak, D., Brendel, W., Constantinidis, C., Feierstein, C. E., Kepecs, A., Mainen, Z. F., ... & Machens, C. K. (2016). Demixed principal component analysis of neural population data. elife, 5, e10989.
>
> [5] Balasubramanian, M., & Schwartz, E. L. (2002). The isomap algorithm and topological stability. Science, 295(5552), 7-7.
>
> [6] Fortunato, C., Bennasar-Vázquez, J., Park, J., Chang, J. C., Miller, L. E., Dudman, J. T., ... & Gallego, J. A. (2024). Nonlinear manifolds underlie neural population activity during behaviour. bioRxiv, 2023-07.
>
> **Trials**: Great catch, we have removed the second ‘trials’ as follows:
>
> >“Sessions were grouped by year, and trials were labeled by the direction…”
>
> **Supplementary references**: We have added references to the supplementary materials in the following sections:
>
> In Section 4.2:
> >“…We calculated the preferred tuning vector of each channel for each session by fitting independent linear regressions between each channel and the position of both degrees of freedom, and taking the weights of each as a vector, as described in Appendix C.2. These vectors are then decomposed into a ’tuning direction’ and ’tuning strength’, as shown in Figure 3A. Figure 3B visualizes the preferred tunings of two channels over the entire dataset,...”
>
> In Section 4.2 (same paragraph):
> >“To quantify the range of tuning angles per channel across time, we measured the circular median and circular IQR of tuning angle per channel over the whole dataset and the median and IQR of tuning strength per channel over the whole dataset (Figure 3E, described in Appendix C.2)...”
>
> In Section 4.3:
> >“The SBP of the training data was z-scored and used to train a ridge regression (RR) and a long short-term memory network (LSTM, [68]). More details on decoder training can be found in Appendix C.1. These decoders…”
>
> **Unclear sentence in Section 5**: Great question. Re-reading it, the last half of this paragraph was very unclear. Here’s an updated version with clarifications:
>
> >“...available in the neural data. When testing decoders on past and future data, we noted that both RR and LSTM exhibited sharp decreases in performance over the short-term, agreeing with previous observations [31, 25]. On longer timescales, however, this decrease stabilizes, likely because the tuning directions of many channels does not change much over the dataset. This would also explain why the shape of the decay was agnostic to the decoder used. Regardless, there seems to be a highly-stable subspace of neural activity that is at least partially behaviorally-relevant and nonlinearly related to behavior.”

---

> > ### Comment · Reviewer_8BwM · 2025-08-04
> >
> > I acknowledge the author’s helpful answers to my concerns; I will keep my rating consistent.

---

### Official Review · Reviewer_HR1R · 2025-07-03

**Rating:** 4
**Confidence:** 4

**Summary:**

The authors presented the LINK Dataset (Long-term Intracortical Neural activity and Kinematics), which contains intracortical spiking activity and kinematic data from 312 sessions of a non-human primate performing a dexterous, two degree-of-freedom finger movement task, spanning 1,242 days.

**Dataset Code Accessibility:**

Yes

**Ethical Considerations:**

No, there are no or only very minor ethics concerns

**Limitations Weaknesses:**

Comments:

In Section 3.1, the authors stated: “Monkey N’s left hand was positioned in a manipulandum...” However, the figure appears to show the right hand. Please clarify or correct the inconsistency.

For Figures 2, 3, and 4, it would be helpful to include a concise summary in the captions to guide the reader through the overall purpose and findings of each figure.

In Figure 4, please include the chance-level prediction accuracy. This would provide a useful reference point for interpreting the performance of the regressor or LSTM models.

It would be informative to report the animal's behavioural performance over the four years of data collection. Demonstrating consistency or change in behavioural metrics would strengthen the dataset's utility for studying long-term stability.

**Strengths Contributions:**

Overall, this is a well-structured dataset with clear documentation, and the limitations are appropriately addressed. As the authors noted, this is the longest set of neural recordings from a Utah array ever released, making it a significant contribution to the brain–machine interface (BMI) field, particularly for researchers investigating long-term algorithmic stability.

---

> ### Author Rebuttal · Authors · 2025-07-30
>
> Thank you for the helpful comments, which we address below:
>
> **Section 1.3**: Thanks for catching that, we have modified the Figure 1’s caption to clarify:
>
> >“…positions (using flex sensors) of the two finger groups, index and middle-ring-small. Note that the illustration depicts Monkey N performing the task with his right hand, but in reality he performed the task with his left.”
>
> **Figs. 2-4 summaries**: Great suggestion, we have added a summary to the beginning of the captions for Figures 2, 3, and 4. See these additions below:
>
> In Figure 2:
> >“Figure 2: Changes in neural signals over time. Here we investigated the change in neural activity over time. Additionally, we briefly examine how population-level neural activity relates to behavior over the course of the dataset. A. In this plot…”
>
> In Figure 3:
> >“Figure 3: Single-channel preferred tuning over time. We investigated the relationship between the SBP of individual neural channels to behavior over the entire dataset by measuring the preferred tuning vectors of each channel for every session. In general, channels tended to stay tuned to the same direction throughout the dataset with varying tuning strengths. A. Toy example…”
>
> In Figure 4:
> >“Figure 4: Baseline decoding performance over the whole dataset. Prediction accuracy of two baseline decoders (RR and LSTM) did not decline over the course of the dataset. However, neither generalized well when testing on past or future sessions (relative to when they were trained). Each session was split into a 300-trial training dataset and 75-trial hold-out…”
>
> **Fig. 4 chance-level accuracy**: We agree with the sentiment that some kind of randomization would provide intuition on how well a decoder is performing. However, since this is a continuous regression problem, chance-level prediction accuracy is not as easily applicable. Instead, we tested decoders trained on a single day on randomly shuffled neural/behavioral data, and added this result to Appendix C.1:
>
> >“…on a 0-1 (and sometimes negative) scale. As a reference point for interpreting model performance of the RR and LSTM models, in one session, we independently shuffled the neural and behavioral data of the holdout set and predicted behavior using  RR and LSTM models trained on that day. We then measured the average R2 of these predictions with the shuffled behavior data. RR had an average R2 of -0.166, and LSTM had an average R2 of -0.156.”
>
> **Behavioral performance**: Great idea! We added Figure S2 to the supplemental materials. While we can’t show it here, we’ve included the caption:
>
> >“Figure S2: Average trial throughput per day. Throughput (measured in bits per second, Fitt’s Law [1]) was calculated per trial. Here, the black line shows the average throughput across all trials in a single day. The shaded area shows the standard error of the mean.”
>
> In addition, we’ve added the following sentences to Section A.1:
>
> >“Trials were performed in continuous blocks, known as ‘sessions’. We used Fitt’s law to measure per-trial throughput (bits per second), as described in [1] as a metric of real-time performance. In S2, we show the average throughput across all trials for each day. Monkey N’s behavior gradually improved throughout the course of the dataset.”

---

> > ### Comment · Area_Chair_NGEF · 2025-08-05
> >
> > Reviewer HR1R, please respond to the authors' rebuttal as soon as possible. The author–reviewer discussion period is nearing its end.

---

> > ### Comment · Reviewer_HR1R · 2025-08-06
> > **comment**
> >
> > The authors have fully addressed my questions. I will keep my scores and confidence.

---

### Official Review · Reviewer_TZPf · 2025-07-14

**Rating:** 5
**Confidence:** 4

**Summary:**

The work introduces the Long-term Intracortical Neural activity and Kinematics Dataset (LINK), a comprehensive dataset designed to address key challenges in intracortical brain-machine interfaces (iBMIs) and neural population dynamics. A major obstacle in iBMIs is the instability of neural signals over long timescales due to biological factors such as foreign body response and neuronal death, which necessitates frequent recalibration of decoders. While prior datasets like FALCON have provided valuable benchmarks, they remain limited in temporal scope and task diversity, restricting their utility for evaluating long-term domain adaptation methods. To fill these gaps, LINK offers 312 sessions over 303 days (~3.5 years) from a single monkey performing a finger movement task, including preprocessed neural features and kinematics for both decoding and population-level analyses. The dataset is positioned as a benchmark for developing domain adaptation techniques in nonstationary neural and kinematic data, with broader implications for timeseries forecasting in machine learning. Preliminary analyses demonstrate the dataset’s utility in probing neural-behavioral relationships and evaluating decoding performance. By providing long-term, high-resolution neural recordings, LINK enables research into the stability of neural representations, the robustness of adaptive decoders, and the stability of latent neural manifold over time. This resource aligns with the goal for advancing both iBMI technology and domain adaptation methods for timeseries.

**Dataset Code Accessibility:**

Yes

**Dataset Code Comments:**

The dataset is well-structured and available via DANDI Archive, a well-known open dataset platform for neuroscience. The code is available in the supplementary materials.

**Ethical Considerations:**

No, there are no or only very minor ethics concerns

**Final Justification:**

All major issues have been addressed and I am more confident in my review after the rebuttal. Additional thoughts can be found in the comment below and I have raised my score for acceptance.

**Limitations Weaknesses:**

Besides the limitations mentioned by the authors, my primary concern lies in the decoder training methodology. While the authors acknowledge domain adaptation techniques for mitigating data distribution drift, the benchmark does not evaluate such algorithms, or even in approximate settings (for instance, continual learning, which is an active research area). The current protocol requires resetting and retraining models from scratch for each session, which is possibly unrealistic in practice. A more practically relevant approach would involve incremental updates (e.g., training RNNs as new data arrives, possibly retaining old data in a buffer for replaying). Incorporating a continual learning evaluation would significantly strengthen the benchmark, as it aligns better with real-world scenarios where models adapt over time without complete resets. This point significantly impact my evalution regarding this work.

Another point is that the evaluation metric for decoder performance only involves R2 score. However, it remains unclear how this score translates into clinical utility (one of the primary goals of this dataset). For example, if the R2 score for decoder is 0.75 (with max MSE = 0.5 cm), how useful this decoder can be? Would max MSE during trials be a more informed metric? I personally consider max MSE < 0.5 cm might be more informed than R2 scores. But the choice of R2 score does not significantly impact my evaluation regarding this work.

As a final note, this work provides a foundational dataset with immense potential, but the decoder evaluation framework feels slightly underexplored. In its current form, this is a very solid borderline accept case for me; addressing the above concern would elevate its impact. If this dataset turns out to be a great dataset for benchmarking algorithms for continual learning problems or other concrete ML problems, my score will be raised significantly.

**Strengths Contributions:**

This is an exceptional and highly valuable dataset. Long-timespan neural recordings are notoriously difficult and expensive to acquire, yet this dataset surpasses prior efforts in both temporal coverage and task diversity. The validation is thorough, including analyses of neural feature dynamics and tuning direction shifts over time. Notably, decoding performance remains stable when trained on a single session, but interestingly, cross-session generalization degrades proportionally to the temporal distance between sessions, which highlights the challenges of neural distribution shifts.

The data and code are also publicly accessible, which strengthens the reproducibility.

The writing is very clear and the figures are accurate and highly intutitive. I do not have any difficulty in understanding any part of this paper.

---

> ### Author Rebuttal · Authors · 2025-07-30
>
> Thank you for your comments! We address key limitations and concerns below.
>
> **Continual learning framework**: This is a great point. Including basic strategies for improving performance over time would serve as a useful baseline or jumping off point for readers. We anticipate that this dataset will be useful for researchers in continual learning and domain adaptation techniques. To this end, we’ve added two new analyses to the paper: (1) multi-day training and (2) daily fine-tuning on minimal data for each session. We plan to add a new figure, Figure 5, to highlight how these impact performance when testing a decoder on future sessions, as in Figure 4D. In Figure 5A, we will present the average dropoff in decoder performance over 100 days (across all sessions) when trained on up to 4 previous sessions of data (for a total of 5). Here is a preliminary version of the relevant text we will add to Section 4.3:
>
> >"Many state of the art BCI decoding algorithms use multiple days of pre-training to improve performance and generalization. As an initial investigation into the impact of multi-day pre-training on BCI decoding, for each day we trained additional instances of an LSTM model using up to 4 additional past sessions of data (up to 5 sessions total). We excluded from this analysis days that had multiple sessions and the first 10 sessions of the dataset. Additionally, we note that since sessions are not evenly distributed through time, the time span covered by the training sessions does vary. Each multi-day model was then evaluated on all sessions up to 180 days following the most recent training session. Evaluations were grouped by the amount of training data and days following training. Average performance in R2 of each group is shown in Figure 5A. We observed that multi-day training reduces the initial sharp decrease in performance (within ~5 days of training), resulting in a net increase compared to single-day trained models, but this does not seem to impact the longer-term decay in performance. Additionally, day 0 performance decreases slightly, which is to be expected.
>
> >To quantify the changes in prediction accuracy relative to training on only one session, we fit a double-exponential decay to each condition (single day of training, or multi-day training) over 180 days in the dataset. The steepest decay, observed during the initial time period, shows the greatest difference: lambda_1 = -1.43 in the single day of training case, and -1.03 in the multi-day training case. The second decay, observed over longer time scales, were comparable: lambda_2 = -0.02 (single day) vs -0.01 (multi-day)."
>
>
> In Figure 5B, we will present the average dropoff in decoder performance over 100 days (across 20 seed days) when an LSTM trained on Day 0 is fine-tuned on up to 5 minutes of data from a subsequent Day K. Here is a preliminary version of the relevant text we will add to Section 4.3:
>
> >"A complementary approach to pretraining is to use continual learning to fine-tune the decoder on small amounts of new-day data, gradually adapting the decoder to shifting neural signals. This removes the need to collect a full set of training data every day. To simulate this, we trained an LSTM on a seed day, and then, for up to 100 days following training, we fine-tuned the LSTM on each consecutive session using 30 s to 5 minutes of training data. We repeated this for 20 seed days approximately evenly spaced across the dataset. As shown in Figure 5B, fine-tuning with only 30 s of data helped to level out the longer-term decay in accuracy, and long term accuracy approached day-0 accuracy as more fine-tuning data was used. We quantified these changes in accuracy using 1 minute of fine-tuning data. At 0 days, decoding R2 was 0.573. For 5, 10, 50, and 100 days post initial training, the R2 was 0.497, 0.452, 0.507 and 0.503 respectively. Thus, fine-tuning with small amounts of labelled data can help maintain accuracy."
>
>
> **Decoder performance metrics**: Thank you for the comment, this is an often discussed issue in the field. We have added a few sentences to the ‘Decoder Evaluation’ paragraph in section C.1 to clarify:
>
> >“…kinematics for each of the four kinematic outputs. All decoders were evaluated by measuring the coefficient of determination (R2) as a measure of prediction accuracy. R2 was chosen over MSE because MSE would need to be evaluated separately for velocity and position as they have different units and often different orders of magnitude. Additionally, these units are not as intuitive in the context of our task (e.g. velocity is defined as the proportion of flexion/20 ms bins). R2, on the other hand, provides a more interpretable metric on a 0-1 (and sometimes negative) scale that can generalize across tasks. As a reference point…”

---

> > ### Comment · Reviewer_TZPf · 2025-08-02
> > **Thanks for the rebuttal and additional thoughts**
> >
> > I thank and applaud all authors for their professional replies regarding all reviewers' comments within limited time. All my questions have been addressed and I'm more confident in my accessment and I will raise my score to 5 (i.e. accept).
> >
> > On another note, I later realized there are subtle differences between the neural decoding here and standard continual learning problems, because here new input neural data may not always come with its new supervisory labels (whereas in some continual learning problems the labels always come with the data but the mapping relations shift over time). Maybe certain amount of unsupervised/semi-supervised learning can be leveraged for the ML/DL communities to better solve this.
> >
> > $\textbf{I would like to applaud the authors again}$ for their efforts to best approximate the continual learning framework by introducing multi-day training and fine-tuning evaluations, which makes the decoder evaluation methodology much more practical.
> >
> > Regarding the newly included results on multi-day training and fine-tuning evaluations, it has demonstrated that basic methods mentioned above for countering neural drifts are helpful yet still have limitations (the performance can degrade over time). Therefore, future algorithms that are designed for better solving this challenge can be benchmarked. This has addressed the most critical issue I mentioned in my initial review.
> >
> > Regarding R2 score, I understand the points raised by the authors but I would be curious to see some concrete numbers in the final supplementary materials if time still permits.
> >
> > Regarding the reviews from Reviewer 1g5E, I suspect there might be certain amount of hallucinations caused by non-human agents and I will support the authors to report this matter via appropriate channels.

---

> > > ### Comment · Reviewer_TZPf · 2025-08-02
> > > **Another remark**
> > >
> > > "At 0 days, decoding R2 was 0.573. For 5, 10, 50, and 100 days post initial training, the R2 was 0.497, 0.452, 0.507 and 0.503 respectively."
> > >
> > > It is interesting to see the performance jumps up and down despite different delays post-training, but it is reasonable since neural representation tragetory can move back and forth as well.

---

> > > > ### Comment · Reviewer_TZPf · 2025-08-07
> > > > **Comments regarding pre processing**
> > > >
> > > > I have noticed some comments regarding the limitation that the dataset doesn't include raw voltage trace. I would like to add a few sentences to defend the authors because raw voltage data can easily go up to hundreds of TB or even more, for long timespan. Massive downsampling and preprocessing are almost always required in order to make data managable. I personally think there's not too much to complain here.

---

### Official Review · Reviewer_1g5E · 2025-07-20

[review text omitted: it was posted to a different submission]

---

> ### Author Rebuttal · Authors · 2025-07-30
>
> Thank you for your review. It is our belief that the majority of the statements made in this review are incorrect, or were intended for a different paper. The following points summarize a few of the discrepancies we noticed:
>
> In the first two sentences of the review, the reviewer states that the dataset comprises data from multiple human participants in the BrainGate clinical trial. This dataset contains data from one able-bodied non-human primate (Macaca mulatta), not 7 human participants with tetraplegia (Homo sapiens sapiens). Additionally, we are not affiliated with the BrainGate clinical trial. Please refer to the abstract for additional details.
>
> The review states that the dataset contains 2400 hours of data across 295 sessions. We do not report the size of the dataset as a measure of time, and the dataset contains 312 sessions, not 295. A rough estimate would be more like ~50 hours.
>
> Twice, the review praises the dataset for including both raw/broadband data and preprocessed neural features in the dataset. The review then points out that, for ease-of-use, we should add preprocessed neural features to the dataset, rather than only including raw/broadband data.The LINK dataset does not include raw/broadband neural data, only preprocessed neural features. Please refer to Figure 1 and Section 3.1 for details.
>
> The review states that a significant portion of the dataset lacks synchronized behavior labels. As stated in the abstract, continuous behavioral data is available for all 312 sessions.
>
> The review states that the paper has a figure outlining the dataset structure, and that Figure 1 illustrates dataset composition. The dataset structure/format is outlined in Section B.3, and Figure 1 is more of a general summary of the methods, including experimental setup and neural feature extraction.
>
> The review states: “Although the paper includes some descriptive analyses and visualizations (e.g., Figure 4), there is minimal demonstration of modeling tasks or performance benchmarks.” Figure 4 presents a solid demonstration of modeling tasks and simple decoder performance across the timescale supplied by our dataset.
>
> The reviewer notes that the dataset has a limited scope for generalization due to the fact that all seven (human) participants are tetraplegic and have Utah array implants. Again, this is not a human dataset. Note that implanting able-bodied human participants with non-FDA approved devices would introduce several “scope-limiting” clinical, legal, and ethical complications.

---

> > ### Comment · Area_Chair_NGEF · 2025-08-04
> >
> > Dear authors,
> >
> > Please be assured that I’m closely following this case of an apparent LLM-generated review. The review has already been flagged and will be further escalated.
> >
> > AC

---

### Decision · Program_Chairs · 2025-09-18

**Decision:**

Accept (poster)

**Comment:**

This work presents longitudinal neural recordings from the motor cortex of a macaque performing a finger movement task, collected over 312 sessions across 3.5 years. Reviewers agreed that this dataset may represent a milestone in invasive BCI development, as it offers an unusually broad view of signal non-stationarity arising from both neuroplasticity and scar-tissue formation. However, the dataset has inherent limitations, namely the use of a single subject and the availability of only pre-processed features, which tempered reviewers' enthusiasm. Excluding one reviewer disqualified for submitting an LLM-generated review, the remaining reviewers were divided between "accept" and "borderline accept" (5, 5, 4, 4). Given the potential impact of this work on neuroprosthesis development, I recommend acceptance.